# Higher-Order Factorization Machines

**Mathieu Blondel, Akinori Fujino, Naonori Ueda**
NTT Communication Science Laboratories
Japan

**Masakazu Ishihata**
Hokkaido University
Japan

## Abstract

Factorization machines (FMs) are a supervised learning approach that can use second-order feature combinations even when the data is very high-dimensional. Unfortunately, despite increasing interest in FMs, there exists to date no efficient training algorithm for higher-order FMs (HOFMs). In this paper, we present the first generic yet efficient algorithms for training arbitrary-order HOFMs. We also present new variants of HOFMs with shared parameters, which greatly reduce model size and prediction times while maintaining similar accuracy. We demonstrate the proposed approaches on four different link prediction tasks.

## 1   Introduction

Factorization machines (FMs) [13, 14] are a supervised learning approach that can use second-order feature combinations efficiently even when the data is very high-dimensional. The key idea of FMs is to model the weights of feature combinations using a *low-rank* matrix. This has two main benefits. First, FMs can achieve empirical accuracy on a par with polynomial regression or kernel methods but with smaller and faster to evaluate models [4]. Second, FMs can infer the weights of feature combinations that were not observed in the training set. This second property is crucial for instance in recommender systems, a domain where FMs have become increasingly popular [14, 16]. Without the low-rank property, FMs would fail to generalize to unseen user-item interactions.

Unfortunately, although higher-order FMs (HOFMs) were briefly mentioned in the original work of [13, 14], there exists to date no efficient algorithm for training arbitrary-order HOFMs. In fact, even just computing predictions given the model parameters naively takes polynomial time in the number of features. For this reason, HOFMs have, to our knowledge, never been applied to any problem. In addition, HOFMs, as originally defined in [13, 14], model each degree in the polynomial expansion with a different matrix and therefore require the estimation of a large number of parameters.

In this paper, we propose the first efficient algorithms for training arbitrary-order HOFMs. To do so, we rely on a link between FMs and the so-called ANOVA kernel [4]. We propose linear-time dynamic programming algorithms for evaluating the ANOVA kernel and computing its gradient. Based on these, we propose stochastic gradient and coordinate descent algorithms for arbitrary-order HOFMs. To reduce the number of parameters, as well as prediction times, we also introduce two new kernels derived from the ANOVA kernel, allowing us to define new variants of HOFMs with shared parameters. We demonstrate the proposed approaches on four different link prediction tasks.

## 2   Factorization machines (FMs)

**Second-order FMs.** Factorization machines (FMs) [13, 14] are an increasingly popular method for efficiently using second-order feature combinations in classification or regression tasks even when the data is very high-dimensional. Let $\boldsymbol{w} \in \mathbb{R}^d$ and $\boldsymbol{P} \in \mathbb{R}^{d \times k}$, where $k \in \mathbb{N}$ is a rank hyper-parameter. We denote the rows of $\boldsymbol{P}$ by $\bar{\boldsymbol{p}}_j$ and its columns by $\boldsymbol{p}_s$, for $j \in [d]$ and $s \in [k]$,

where $[d] := \{1, \ldots, d\}$. Then, FMs predict an output $y \in \mathbb{R}$ from a vector $\boldsymbol{x} = [x_1, \ldots, x_d]^{\mathrm{T}}$ by

$$\hat{y}_{\mathrm{FM}}(\boldsymbol{x}) := \langle \boldsymbol{w}, \boldsymbol{x} \rangle + \sum_{j' > j} \langle \bar{\boldsymbol{p}}_j, \bar{\boldsymbol{p}}_{j'} \rangle x_j x_{j'}. \tag{1}$$

An important characteristic of (1) is that it considers only combinations of *distinct features* (i.e., the squared features $x_1^2, \ldots, x_d^2$ are ignored). The main advantage of FMs compared to naive polynomial regression is that the number of parameters to estimate is $O(dk)$ instead of $O(d^2)$. In addition, we can compute predictions in $O(2dk)$ time[1] using

$$\hat{y}_{\mathrm{FM}}(\boldsymbol{x}) = \boldsymbol{w}^{\mathrm{T}} \boldsymbol{x} + \frac{1}{2} \left( \|\boldsymbol{P}^{\mathrm{T}} \boldsymbol{x}\|^2 - \sum_{s=1}^{k} \|\boldsymbol{p}_s \circ \boldsymbol{x}\|^2 \right),$$

where $\circ$ indicates element-wise product [3]. Given a training set $\boldsymbol{X} = [\boldsymbol{x}_1, \ldots, \boldsymbol{x}_n] \in \mathbb{R}^{d \times n}$ and $\boldsymbol{y} = [y_1, \ldots, y_n]^{\mathrm{T}} \in \mathbb{R}^n$, $\boldsymbol{w}$ and $\boldsymbol{P}$ can be learned by minimizing the following non-convex objective

$$\frac{1}{n} \sum_{i=1}^{n} \ell(y_i, \hat{y}_{\mathrm{FM}}(\boldsymbol{x}_i)) + \frac{\beta_1}{2} \|\boldsymbol{w}\|^2 + \frac{\beta_2}{2} \|\boldsymbol{P}\|^2, \tag{2}$$

where $\ell$ is a convex loss function and $\beta_1 > 0, \beta_2 > 0$ are hyper-parameters. The popular `libfm` library [14] implements efficient stochastic gradient and coordinate descent algorithms for obtaining a stationary point of (2). Both algorithms have a runtime complexity of $O(2dkn)$ per epoch.

**Higher-order FMs (HOFMs).** Although no training algorithm was provided, FMs were extended to higher-order feature combinations in the original work of [13, 14]. Let $\boldsymbol{P}^{(t)} \in \mathbb{R}^{d \times k_t}$, where $t \in \{2, \ldots, m\}$ is the order or degree of feature combinations considered, and $k_t \in \mathbb{N}$ is a rank hyper-parameter. Let $\bar{\boldsymbol{p}}_j^{(t)}$ be the $j^{\mathrm{th}}$ row of $\boldsymbol{P}^{(t)}$. Then $m$-order HOFMs can be defined as

$$\hat{y}_{\mathrm{HOFM}}(\boldsymbol{x}) := \langle \boldsymbol{w}, \boldsymbol{x} \rangle + \sum_{j' > j} \langle \bar{\boldsymbol{p}}_j^{(2)}, \bar{\boldsymbol{p}}_{j'}^{(2)} \rangle x_j x_{j'} + \cdots + \sum_{j_m > \cdots > j_1} \langle \bar{\boldsymbol{p}}_{j_1}^{(m)}, \ldots, \bar{\boldsymbol{p}}_{j_m}^{(m)} \rangle x_{j_1} x_{j_2} \ldots x_{j_m} \tag{3}$$

where we defined $\langle \bar{\boldsymbol{p}}_{j_1}^{(t)}, \ldots, \bar{\boldsymbol{p}}_{j_t}^{(t)} \rangle := \mathrm{sum}(\bar{\boldsymbol{p}}_{j_1}^{(t)} \circ \cdots \circ \bar{\boldsymbol{p}}_{j_t}^{(t)})$ (sum of element-wise products). The objective function of HOFMs can be expressed in a similar way as for (2):

$$\frac{1}{n} \sum_{i=1}^{n} \ell(y_i, \hat{y}_{\mathrm{HOFM}}(\boldsymbol{x}_i)) + \frac{\beta_1}{2} \|\boldsymbol{w}\|^2 + \sum_{t=2}^{m} \frac{\beta_t}{2} \|\boldsymbol{P}^{(t)}\|^2, \tag{4}$$

where $\beta_1, \ldots, \beta_m > 0$ are hyper-parameters. To avoid the combinatorial explosion of hyper-parameter combinations to search, in our experiments we will simply set $\beta_1 = \cdots = \beta_m$ and $k_2 = \cdots = k_m$. While (3) looks quite daunting, [4] recently showed that FMs can be expressed from a simpler kernel perspective. Let us define the ANOVA[2] kernel [19] of degree $2 \le m \le d$ by

$$\mathcal{A}^m(\boldsymbol{p}, \boldsymbol{x}) := \sum_{j_m > \cdots > j_1} \prod_{t=1}^{m} p_{j_t} x_{j_t}. \tag{5}$$

For later convenience, we also define $\mathcal{A}^0(\boldsymbol{p}, \boldsymbol{x}) := 1$ and $\mathcal{A}^1(\boldsymbol{p}, \boldsymbol{x}) := \langle \boldsymbol{p}, \boldsymbol{x} \rangle$. Then it is shown that

$$\hat{y}_{\mathrm{HOFM}}(\boldsymbol{x}) = \langle \boldsymbol{w}, \boldsymbol{x} \rangle + \sum_{s=1}^{k_2} \mathcal{A}^2 \left( \boldsymbol{p}_s^{(2)}, \boldsymbol{x} \right) + \cdots + \sum_{s=1}^{k_m} \mathcal{A}^m \left( \boldsymbol{p}_s^{(m)}, \boldsymbol{x} \right), \tag{6}$$

where $\boldsymbol{p}_s^{(t)}$ is the $s^{\mathrm{th}}$ column of $\boldsymbol{P}^{(t)}$. This perspective shows that we can view FMs and HOFMs as a type of kernel machine whose "support vectors" are learned directly from data. Intuitively, the ANOVA kernel can be thought as a kind of polynomial kernel that uses feature combinations without replacement (i.e., of *distinct* features). A key property of the ANOVA kernel is multi-linearity [4]:

$$\mathcal{A}^m(\boldsymbol{p}, \boldsymbol{x}) = \mathcal{A}^m(\boldsymbol{p}_{\neg j}, \boldsymbol{x}_{\neg j}) + p_j x_j \, \mathcal{A}^{m-1}(\boldsymbol{p}_{\neg j}, \boldsymbol{x}_{\neg j}), \tag{7}$$

where $\boldsymbol{p}_{\neg j}$ denotes the $(d-1)$-dimensional vector with $p_j$ removed and similarly for $\boldsymbol{x}_{\neg j}$. That is, everything else kept fixed, $\mathcal{A}^m(\boldsymbol{p}, \boldsymbol{x})$ is an affine function of $p_j \; \forall j \in [d]$. Although no training

algorithm was provided, [4] showed based on (7) that, although non-convex, the objective function of arbitrary-order HOFMs is convex in $\boldsymbol{w}$ and in each row of $\boldsymbol{P}^{(2)}, \ldots, \boldsymbol{P}^{(m)}$, separately.

**Interpretability of HOFMs.** An advantage of FMs and HOFMs is their interpretability. To see why this is the case, notice that we can rewrite (3) as

$$\hat{y}_{\text{HOFM}}(\boldsymbol{x}) = \langle \boldsymbol{w}, \boldsymbol{x} \rangle + \sum_{j'>j} \boldsymbol{\mathcal{W}}_{j,j'}^{(2)} x_j x_{j'} + \cdots + \sum_{j_m > \cdots > j_1} \boldsymbol{\mathcal{W}}_{j_1,\ldots,j_m}^{(m)} x_{j_1} x_{j_2} \ldots x_{j_m},$$

where we defined $\boldsymbol{\mathcal{W}}^{(t)} := \sum_{s=1}^{k_t} \underbrace{\boldsymbol{p}_s^{(t)} \otimes \cdots \otimes \boldsymbol{p}_s^{(t)}}_{t \text{ times}}$. Intuitively, $\boldsymbol{\mathcal{W}}^{(t)} \in \mathbb{R}^{d^t}$ is a low-rank $t$-way tensor which contains the weights of feature combinations of degree $t$. For instance, when $t = 3$, $\boldsymbol{\mathcal{W}}_{i,j,k}^{(3)}$ is the weight of $x_i x_j x_k$. Similarly to the ANOVA decomposition of functions, HOFMs consider only combinations of distinct features (i.e., $x_{j_1} x_{j_2} \ldots x_{j_m}$ for $j_m > \cdots > j_2 > j_1$).

**This paper.** Unfortunately, there exists to date no efficient algorithm for training arbitrary-order HOFMs. Indeed, computing (5) naively takes $O(d^m)$, i.e., polynomial time. In the following, we present linear-time algorithms. Moreover, HOFMs, as originally defined in [13, 14] require the estimation of $m-1$ matrices $\boldsymbol{P}^{(2)}, \ldots, \boldsymbol{P}^{(m)}$. Thus, HOFMs can produce large models when $m$ is large. To address this issue, we propose new variants of HOFMs with shared parameters.

## 3 Linear-time stochastic gradient algorithms for HOFMs

The kernel view presented in Section 2 allows us to focus on the ANOVA kernel as the main "computational unit" for training HOFMs. In this section, we develop dynamic programming (DP) algorithms for evaluating the ANOVA kernel and computing its gradient in only $O(dm)$ time.

**Evaluation.** The main observation (see also [18, Section 9.2]) is that we can use (7) to recursively remove features until computing the kernel becomes trivial. Let us denote a subvector of $\boldsymbol{p}$ by $\boldsymbol{p}_{1:j} \in \mathbb{R}^j$ and similarly for $\boldsymbol{x}$. Let us introduce the shorthand $a_{j,t} := \mathcal{A}^t(\boldsymbol{p}_{1:j}, \boldsymbol{x}_{1:j})$. Then, from (7),

$$a_{j,t} = a_{j-1,t} + p_j x_j \, a_{j-1,t-1} \quad \forall d \geq j \geq t \geq 1. \tag{8}$$

For convenience, we also define $a_{j,0} = 1 \, \forall j \geq 0$ since $\mathcal{A}^0(\boldsymbol{p}, \boldsymbol{x}) = 1$ and $a_{j,t} = 0 \, \forall j < t$ since there does not exist any $t$-combination of features in a $j < t$ dimensional vector.

The quantity we want to compute is $\mathcal{A}^m(\boldsymbol{p}, \boldsymbol{x}) = a_{d,m}$. Instead of naively using recursion (8), which would lead to many redundant computations, we use a bottom-up approach and organize computations in a DP table. We start from the top-left corner to initialize the recursion and go through the table to arrive at the solution in the bottom-right corner. The procedure, summarized in Algorithm 1, takes $O(dm)$ time and memory.

Table 1: Example of DP table

|       | $j = 0$ | $j = 1$   | $j = 2$   | $\ldots$ | $j = d$   |
|-------|---------|-----------|-----------|----------|-----------|
| $t = 0$ | 1       | 1         | 1         | 1        | 1         |
| $t = 1$ | 0       | $a_{1,1}$ | $a_{2,1}$ | $\ldots$ | $a_{d,1}$ |
| $t = 2$ | 0       | 0         | $a_{2,2}$ | $\ldots$ | $a_{d,2}$ |
| $\vdots$ | $\vdots$ | $\vdots$ | $\vdots$ | $\ddots$ | $\vdots$ |
| $t = m$ | 0       | 0         | 0         | $\ldots$ | $a_{d,m}$ |

**Gradients.** For computing the gradient of $\mathcal{A}^m(\boldsymbol{p}, \boldsymbol{x})$ w.r.t. $\boldsymbol{p}$, we use reverse-mode differentiation [2] (a.k.a. backpropagation in a neural network context), since it allows us to compute the entire gradient in a single pass. We supplement each variable $a_{j,t}$ in the DP table by a so-called adjoint $\tilde{a}_{j,t} := \frac{\partial a_{d,m}}{\partial a_{j,t}}$, which represents the sensitivity of $a_{d,m} = \mathcal{A}^m(\boldsymbol{p}, \boldsymbol{x})$ w.r.t. $a_{j,t}$. From recursion (8), except for edge cases, $a_{j,t}$ influences $a_{j+1,t+1}$ and $a_{j+1,t}$. Using the chain rule, we then obtain

$$\tilde{a}_{j,t} = \frac{\partial a_{d,m}}{\partial a_{j+1,t}} \frac{\partial a_{j+1,t}}{\partial a_{j,t}} + \frac{\partial a_{d,m}}{\partial a_{j+1,t+1}} \frac{\partial a_{j+1,t+1}}{\partial a_{j,t}} = \tilde{a}_{j+1,t} + p_{j+1} x_{j+1} \, \tilde{a}_{j+1,t+1} \quad \forall d-1 \geq j \geq t \geq 1. \tag{9}$$

Similarly, we introduce the adjoint $\tilde{p}_j := \frac{\partial a_{d,m}}{\partial p_j} \, \forall j \in [d]$. Since $p_j$ influences $a_{j,t} \, \forall t \in [m]$, we have

$$\tilde{p}_j = \sum_{t=1}^m \frac{\partial a_{d,m}}{\partial a_{j,t}} \frac{\partial a_{j,t}}{\partial p_j} = \sum_{t=1}^m \tilde{a}_{j,t} \, a_{j-1,t-1} \, x_j.$$

We can run recursion (9) in reverse order of the DP table starting from $\tilde{a}_{d,m} = \frac{\partial a_{d,m}}{\partial a_{d,m}} = 1$. Using this approach, we can compute the entire gradient $\nabla \mathcal{A}^m(\boldsymbol{p}, \boldsymbol{x}) = [\tilde{p}_1, \ldots, \tilde{p}_d]^{\text{T}}$ w.r.t. $\boldsymbol{p}$ in $O(dm)$ time and memory. The procedure is summarized in Algorithm 2.

| **Algorithm 1** Evaluating $\mathcal{A}^m(\boldsymbol{p}, \boldsymbol{x})$ in $O(dm)$ | **Algorithm 2** Computing $\nabla \mathcal{A}^m(\boldsymbol{p}, \boldsymbol{x})$ in $O(dm)$ |
|---|---|
| **Input:** $\boldsymbol{p} \in \mathbb{R}^d$, $\boldsymbol{x} \in \mathbb{R}^d$ | **Input:** $\boldsymbol{p} \in \mathbb{R}^d$, $\boldsymbol{x} \in \mathbb{R}^d$, $\{a_{j,t}\}_{j,t=0}^{d,m}$ |
| $a_{j,t} \leftarrow 0 \ \forall t \in [m], j \in [d] \cup \{0\}$ | $\tilde{a}_{j,t} \leftarrow 0 \ \forall t \in [m+1], j \in [d]$ |
| $a_{j,0} \leftarrow 1 \ \forall j \in [d] \cup \{0\}$ | $\tilde{a}_{d,m} \leftarrow 1$ |
| | |
| **for** $t := 1, \ldots, m$ **do** | **for** $t := m, \ldots, 1$ **do** |
|     **for** $j := t, \ldots, d$ **do** |     **for** $j := d-1, \ldots, t$ **do** |
|         $a_{j,t} \leftarrow a_{j-1,t} + p_j x_j a_{j-1,t-1}$ |         $\tilde{a}_{j,t} \leftarrow \tilde{a}_{j+1,t} + \tilde{a}_{j+1,t+1} p_{j+1} x_{j+1}$ |
|     **end for** |     **end for** |
| **end for** | **end for** |
| | |
| | $\tilde{p}_j := \sum_{t=1}^m \tilde{a}_{j,t} a_{j-1,t-1} x_j \ \forall j \in [d]$ |
| **Output:** $\mathcal{A}^m(\boldsymbol{p}, \boldsymbol{x}) = a_{d,m}$ | **Output:** $\nabla \mathcal{A}^m(\boldsymbol{p}, \boldsymbol{x}) = [\tilde{p}_1, \ldots, \tilde{p}_d]^{\mathrm{T}}$ |

**Stochastic gradient (SG) algorithms.** Based on Algorithm 1 and 2, we can easily learn arbitrary-order HOFMs using any gradient-based optimization algorithm. Here we focus our discussion on SG algorithms. If we alternatingly minimize (4) w.r.t $\boldsymbol{P}^{(2)}, \ldots, \boldsymbol{P}^{(m)}$, then the sub-problem associated with degree $m$ is of the form

$$F(\boldsymbol{P}) := \frac{1}{n} \sum_{i=1}^n \ell \left( y_i, \sum_{s=1}^k \mathcal{A}^m(\boldsymbol{p}_s, \boldsymbol{x}_i) + o_i \right) + \frac{\beta}{2} \|\boldsymbol{P}\|^2, \tag{10}$$

where $o_1, \ldots, o_n \in \mathbb{R}$ are fixed offsets which account for the contribution of degrees other than $m$ to the predictions. The sub-problem is convex in each row of $\boldsymbol{P}$ [4]. A SG update for (10) w.r.t. $\boldsymbol{p}_s$ for some instance $\boldsymbol{x}_i$ can be computed by $\boldsymbol{p}_s \leftarrow \boldsymbol{p}_s - \eta \ell'(y_i, \hat{y}_i) \nabla \mathcal{A}^m(\boldsymbol{p}_s, \boldsymbol{x}_i) - \eta \beta \boldsymbol{p}_s$, where $\eta$ is a learning rate and where we defined $\hat{y}_i := \sum_{s=1}^k \mathcal{A}^m(\boldsymbol{p}_s, \boldsymbol{x}_i) + o_i$. Because evaluating $\mathcal{A}^m(\boldsymbol{p}, \boldsymbol{x})$ and computing its gradient both take $O(dm)$, the cost per epoch, i.e., of visiting all instances, is $O(mdkn)$. When $m = 2$, this is the same cost as the SG algorithm implemented in `libfm`.

**Sparse data.** We conclude this section with a few useful remarks on sparse data. Let us denote the support of a vector $\boldsymbol{x} = [x_1, \ldots, x_d]^{\mathrm{T}}$ by $\mathrm{supp}(\boldsymbol{x}) := \{j \in [d]: x_j \neq 0\}$ and let us define $\boldsymbol{x}_S := [x_j: j \in S]^{\mathrm{T}}$. It is easy to see from (7) that the gradient and $\boldsymbol{x}$ have the same support, i.e., $\mathrm{supp}(\nabla \mathcal{A}^m(\boldsymbol{p}, \boldsymbol{x})) = \mathrm{supp}(\boldsymbol{x})$. Another useful remark is that $\mathcal{A}^m(\boldsymbol{p}, \boldsymbol{x}) = \mathcal{A}^m(\boldsymbol{p}_{\mathrm{supp}(\boldsymbol{x})}, \boldsymbol{x}_{\mathrm{supp}(\boldsymbol{x})})$, provided that $m \leq n_z(\boldsymbol{x})$, where $n_z(\boldsymbol{x})$ is the number of non-zero elements in $\boldsymbol{x}$. Hence, when the data is sparse, we only need to iterate over non-zero features in Algorithm 1 and 2. Consequently, their time and memory cost is only $O(n_z(\boldsymbol{x})m)$ and thus the cost per epoch of SG algorithms is $O(mkn_z(\boldsymbol{X}))$.

## 4 Coordinate descent algorithm for arbitrary-order HOFMs

We now describe a coordinate descent (CD) solver for arbitrary-order HOFMs. CD is a good choice for learning HOFMs because their objective function is coordinate-wise convex, thanks to the multi-linearity of the ANOVA kernel [4]. Our algorithm can be seen as a generalization to higher orders of the CD algorithms proposed in [14, 4].

**An alternative recursion.** Efficient CD implementations typically require maintaining statistics for each training instance, such as the predictions at the current iteration. When a coordinate is updated, the statistics then need to be synchronized. Unfortunately, the recursion we used in the previous section is not suitable for a CD algorithm because it would require to store and synchronize the DP table for each training instance upon coordinate-wise updates. We therefore turn to an alternative recursion:

$$\mathcal{A}^m(\boldsymbol{p}, \boldsymbol{x}) = \frac{1}{m} \sum_{t=1}^m (-1)^{t+1} \mathcal{A}^{m-t}(\boldsymbol{p}, \boldsymbol{x}) \mathcal{D}^t(\boldsymbol{p}, \boldsymbol{x}), \tag{11}$$

where we defined $\mathcal{D}^t(\boldsymbol{p}, \boldsymbol{x}) := \sum_{j=1}^d (p_j x_j)^t$. Note that the recursion was already known in the context of traditional kernel methods (c.f., [19, Section 11.8]) but its application to HOFMs is novel. Since we know that $\mathcal{A}^0(\boldsymbol{p}, \boldsymbol{x}) = 1$ and $\mathcal{A}^1(\boldsymbol{p}, \boldsymbol{x}) = \langle \boldsymbol{p}, \boldsymbol{x} \rangle$, we can use (11) to compute $\mathcal{A}^2(\boldsymbol{p}, \boldsymbol{x})$, then $\mathcal{A}^3(\boldsymbol{p}, \boldsymbol{x})$, and so on. The overall evaluation cost for arbitrary $m \in \mathbb{N}$ is $O(md + m^2)$.

**Coordinate-wise derivatives.** We can apply reverse-mode differentiation to recursion (11) in order to compute the entire gradient (c.f., Appendix C). However, in CD, since we only need the derivative of one variable at a time, we can simply use forward-mode differentiation:

$$\frac{\partial \mathcal{A}^m(\boldsymbol{p}, \boldsymbol{x})}{\partial p_j} = \frac{1}{m} \sum_{t=1}^m (-1)^{t+1} \left[ \frac{\partial \mathcal{A}^{m-t}(\boldsymbol{p}, \boldsymbol{x})}{\partial p_j} \mathcal{D}^t(\boldsymbol{p}, \boldsymbol{x}) + \mathcal{A}^{m-t}(\boldsymbol{p}, \boldsymbol{x}) \frac{\partial \mathcal{D}^t(\boldsymbol{p}, \boldsymbol{x})}{\partial p_j} \right], \qquad (12)$$

where $\frac{\partial \mathcal{D}^t(\boldsymbol{p}, \boldsymbol{x})}{\partial p_j} = t p_j^{t-1} x_j^t$. The advantage of (12) is that we only need to cache $\mathcal{D}^t(\boldsymbol{p}, \boldsymbol{x})$ for $t \in [m]$. Hence the memory complexity per sample is only $O(m)$ instead of $O(dm)$ for (8).

**Use in a CD algorithm.** Similarly to [4], we assume that the loss function $\ell$ is $\mu$-smooth and update the elements $p_{j,s}$ of $\boldsymbol{P}$ in cyclic order by $p_{j,s} \leftarrow p_{j,s} - \eta_{j,s}^{-1} \frac{\partial F(\boldsymbol{P})}{\partial p_{j,s}}$, where we defined

$$\eta_{j,s} := \frac{\mu}{n} \sum_{i=1}^n \left( \frac{\partial \mathcal{A}^m(\boldsymbol{p}_s, \boldsymbol{x}_i)}{\partial p_{j,s}} \right)^2 + \beta \quad \text{and} \quad \frac{\partial F(\boldsymbol{P})}{\partial p_{j,s}} = \frac{1}{n} \sum_{i=1}^n \ell'(y_i, \hat{y}_i) \frac{\partial \mathcal{A}^m(\boldsymbol{p}_s, \boldsymbol{x}_i)}{\partial p_{j,s}} + \beta p_{j,s}.$$

The update guarantees that the objective value is monotonically non-increasing and is the exact coordinate-wise minimizer when $\ell$ is the squared loss. Overall, the total cost per epoch, i.e., updating all coordinates once, is $O(\tau(m) k n_z(\boldsymbol{X}))$, where $\tau(m)$ is the time it takes to compute (12). Assuming $\mathcal{D}^t(\boldsymbol{p}_s, \boldsymbol{x}_i)$ have been previously cached, for $t \in [m]$, computing (12) takes $\tau(m) = m(m+1)/2 - 1$ operations. For fixed $m$, if we unroll the two loops needed to compute (12), modern compilers can often further reduce the number of operations needed. Nevertheless, this quadratic dependency on $m$ means that our CD algorithm is best for small $m$, typically $m \leq 4$.

## 5 HOFMs with shared parameters

HOFMs, as originally defined in [13, 14], model each degree with *separate* matrices $\boldsymbol{P}^{(2)}, \ldots, \boldsymbol{P}^{(m)}$. Assuming that we use the same rank $k$ for all matrices, the total model size of $m$-order HOFMs is therefore $O(kdm)$. Moreover, even when using our $O(dm)$ DP algorithm, the cost of computing predictions is $O(k(2d + \cdots + md)) = O(kdm^2)$. Hence, HOFMs tend to produce large, expensive-to-evaluate models. To reduce model size and prediction times, we introduce two new kernels which allow us to *share* parameters between each degree: the **inhomogeneous ANOVA kernel** and the **all-subsets kernel**. Because both kernels are derived from the ANOVA kernel, they share the same appealing properties: multi-linearity, sparse gradients and sparse-data friendliness.

### 5.1 Inhomogeneous ANOVA kernel

It is well-known that a sum of kernels is equivalent to concatenating their associated feature maps [18, Section 3.4]. Let $\boldsymbol{\theta} = [\theta_1, \ldots, \theta_m]^\mathrm{T}$. To combine different degrees, a natural kernel is therefore

$$\mathcal{A}^{1 \to m}(\boldsymbol{p}, \boldsymbol{x}; \boldsymbol{\theta}) := \sum_{t=1}^m \theta_t \mathcal{A}^t(\boldsymbol{p}, \boldsymbol{x}). \qquad (13)$$

The kernel uses all feature combinations of degrees 1 up to $m$. We call it *inhomogeneous* ANOVA kernel, since it is an inhomogeneous polynomial of $\boldsymbol{x}$. In contrast, $\mathcal{A}^m(\boldsymbol{p}, \boldsymbol{x})$ is homogeneous. The main difference between (13) and (6) is that all ANOVA kernels in the sum share the same parameters. However, to increase modeling power, we allow each kernel to have different weights $\theta_1, \ldots, \theta_m$.

**Evaluation.** Due to the recursive nature of Algorithm 1, when computing $\mathcal{A}^m(\boldsymbol{p}, \boldsymbol{x})$, we also get $\mathcal{A}^1(\boldsymbol{p}, \boldsymbol{x}), \ldots, \mathcal{A}^{m-1}(\boldsymbol{p}, \boldsymbol{x})$ for free. Indeed, lower-degree kernels are available in the last column of the DP table, i.e., $\mathcal{A}^t(\boldsymbol{p}, \boldsymbol{x}) = a_{d,t} \; \forall t \in [m]$. Hence, the cost of evaluating (13) is $O(dm)$ time. The total cost for computing $\hat{y} = \sum_{s=1}^k \mathcal{A}^{1 \to m}(\boldsymbol{p}_s, \boldsymbol{x}; \boldsymbol{\theta})$ is $O(kdm)$ instead of $O(kdm^2)$ for $\hat{y}_{\mathrm{HOFM}}(\boldsymbol{x})$.

**Learning.** While it is certainly possible to learn $\boldsymbol{P}$ and $\boldsymbol{\theta}$ by directly minimizing some objective function, here we propose an easier solution, which works well in practice. Our key observation is that we can easily turn $\mathcal{A}^m$ into $\mathcal{A}^{1 \to m}$ by adding dummy values to feature vectors. Let us denote the concatenation of $\boldsymbol{p}$ with a scalar $\gamma$ by $[\gamma, \boldsymbol{p}]$ and similarly for $\boldsymbol{x}$. From (7), we easily obtain

$$\mathcal{A}^m([\gamma_1, \boldsymbol{p}], [1, \boldsymbol{x}]) = \mathcal{A}^m(\boldsymbol{p}, \boldsymbol{x}) + \gamma_1 \mathcal{A}^{m-1}(\boldsymbol{p}, \boldsymbol{x}).$$

Table 2: Datasets used in our experiments. For a detailed description, c.f. Appendix A.

| Dataset | $n_+$ | Columns of $\boldsymbol{A}$ | $n_A$ | $d_A$ | Columns of $\boldsymbol{B}$ | $n_B$ | $d_B$ |
|---|---|---|---|---|---|---|---|
| NIPS [17] | 4,140 | Authors | 2,037 | 13,649 | | | |
| Enzyme [21] | 2,994 | Enzymes | 668 | 325 | | | |
| GD [10] | 3,954 | Diseases | 3,209 | 3,209 | Genes | 12,331 | 25,275 |
| Movielens 100K [6] | 21,201 | Users | 943 | 49 | Movies | 1,682 | 29 |

Similarly, if we apply (7) twice, we obtain:

$$\mathcal{A}^m([\gamma_1, \gamma_2, \boldsymbol{p}], [1, 1, \boldsymbol{x}]) = \mathcal{A}^m(\boldsymbol{p}, \boldsymbol{x}) + (\gamma_1 + \gamma_2)\mathcal{A}^{m-1}(\boldsymbol{p}, \boldsymbol{x}) + \gamma_1\gamma_2\mathcal{A}^{m-2}(\boldsymbol{p}, \boldsymbol{x}).$$

Applying the above to $m = 2$ and $m = 3$, we obtain

$$\mathcal{A}^2([\gamma_1, \boldsymbol{p}], [1, \boldsymbol{x}]) = \mathcal{A}^{1\to2}(\boldsymbol{p}, \boldsymbol{x}; [\gamma_1, 1]) \quad \text{and} \quad \mathcal{A}^3([\gamma_1, \gamma_2, \boldsymbol{p}], [1, 1, \boldsymbol{x}]) = \mathcal{A}^{1\to3}(\boldsymbol{p}, \boldsymbol{x}; [\gamma_1\gamma_2, \gamma_1+\gamma_2, 1]).$$

More generally, by adding $m-1$ dummy features to $\boldsymbol{p}$ and $\boldsymbol{x}$, we can convert $\mathcal{A}^m$ to $\mathcal{A}^{1\to m}$. Because $\boldsymbol{p}$ is learned, this means that we can automatically learn $\gamma_1, \ldots, \gamma_{m-1}$. These weights can then be converted to $\theta_1, \ldots, \theta_m$ by "unrolling" recursion (7). Although simple, we show in our experiments that this approach works favorably compared to directly learning $\boldsymbol{P}$ and $\boldsymbol{\theta}$. The main advantage of this approach is that we can use the same software unmodified (we simply need to minimize (10) with the augmented data). Moreover, the cost of computing the entire gradient by Algorithm 2 using the augmented data is just $O(dm + m^2)$ compared to $O(dm^2)$ for HOFMs with separate parameters.

### 5.2 All-subsets kernel

We now consider a closely related kernel called all-subsets kernel [18, Definition 9.5]:

$$\mathcal{S}(\boldsymbol{p}, \boldsymbol{x}) := \prod_{j=1}^{d}(1 + p_j x_j).$$

The main difference with the traditional use of this kernel is that we learn $\boldsymbol{p}$. Interestingly, it can be shown that $\mathcal{S}(\boldsymbol{p}, \boldsymbol{x}) = 1 + \mathcal{A}^{1\to d}(\boldsymbol{p}, \boldsymbol{x}; \boldsymbol{1}) = 1 + \mathcal{A}^{1\to n_z(\boldsymbol{x})}(\boldsymbol{p}, \boldsymbol{x}; \boldsymbol{1})$, where $n_z(\boldsymbol{x})$ is the number of non-zero features in $\boldsymbol{x}$. Hence, the kernel uses *all* combinations of distinct features up to order $n_z(\boldsymbol{x})$ *with uniform weights*. Even if $d$ is very large, the kernel can be a good choice if each training instance contains only a few non-zero elements. To learn the parameters, we simply substitute $\mathcal{A}^m$ with $\mathcal{S}$ in (10). In SG or CD algorithms, all it entails is to substitute $\nabla \mathcal{A}^m(\boldsymbol{p}, \boldsymbol{x})$ with $\nabla \mathcal{S}(\boldsymbol{p}, \boldsymbol{x})$. For computing $\nabla \mathcal{S}(\boldsymbol{p}, \boldsymbol{x})$, it is easy to verify that $\mathcal{S}(\boldsymbol{p}, \boldsymbol{x}) = \mathcal{S}(\boldsymbol{p}_{\neg j}, \boldsymbol{x}_{\neg j})(1 + p_j x_j) \, \forall j \in [d]$ and therefore we have

$$\nabla \mathcal{S}(\boldsymbol{p}, \boldsymbol{x}) = \left[ x_1 \, \mathcal{S}(\boldsymbol{p}_{\neg 1}, \boldsymbol{x}_{\neg 1}), \ldots, x_d \, \mathcal{S}(\boldsymbol{p}_{\neg d}, \boldsymbol{x}_{\neg d}) \right]^{\mathrm{T}} = \left[ \frac{x_1 \, \mathcal{S}(\boldsymbol{p}, \boldsymbol{x})}{1 + p_1 x_1}, \ldots, \frac{x_d \, \mathcal{S}(\boldsymbol{p}, \boldsymbol{x})}{1 + p_d x_d} \right]^{\mathrm{T}}.$$

Therefore, the main advantage of the all-subsets kernel is that we can evaluate it and compute its gradient in just $O(d)$ time. The total cost for computing $\hat{y} = \sum_{s=1}^{k} \mathcal{S}(\boldsymbol{p}_s, \boldsymbol{x})$ is only $O(kd)$.

## 6 Experimental results

### 6.1 Application to link prediction

**Problem setting.** We now demonstrate a novel application of HOFMs to predict the presence or absence of links between nodes in a graph. Formally, we assume two sets of possibly disjoint nodes of size $n_A$ and $n_B$, respectively. We assume features for the two sets of nodes, represented by matrices $\boldsymbol{A} \in \mathbb{R}^{d_A \times n_A}$ and $\boldsymbol{B} \in \mathbb{R}^{d_B \times n_B}$. For instance, $\boldsymbol{A}$ can represent user features and $\boldsymbol{B}$ movie features. We denote the columns of $\boldsymbol{A}$ and $\boldsymbol{B}$ by $\boldsymbol{a}_i$ and $\boldsymbol{b}_j$, respectively. We are given a matrix $\boldsymbol{Y} \in \{0, 1\}^{n_A \times n_B}$, whose elements indicate presence (positive sample) or absence (negative sample) of link between two nodes $\boldsymbol{a}_i$ and $\boldsymbol{b}_j$. We denote the number of positive samples by $n_+$. Using this data, our goal is to predict new associations. Datasets used in our experiments are summarized in Table 2. Note that for the NIPS and Enzyme datasets, $\boldsymbol{A} = \boldsymbol{B}$.

**Conversion to a supervised problem.** We need to convert the above information to a format FMs and HOFMs can handle. To predict an element $y_{i,j}$ of $\boldsymbol{Y}$, we simply form $\boldsymbol{x}_{i,j}$ to be the concatenation

Table 3: Comparison of area under the ROC curve (AUC) as measured on the test sets.

| | NIPS | Enzyme | GD | Movielens 100K |
|---|---|---|---|---|
| HOFM ($m = 2$) | 0.856 | 0.880 | 0.717 | 0.778 |
| HOFM ($m = 3$) | **0.875** | **0.888** | 0.717 | 0.786 |
| HOFM ($m = 4$) | 0.874 | 0.887 | 0.717 | 0.786 |
| HOFM ($m = 5$) | 0.874 | 0.887 | 0.717 | 0.786 |
| HOFM-shared-augmented ($m = 2$) | 0.858 | 0.876 | 0.704 | 0.778 |
| HOFM-shared-augmented ($m = 3$) | 0.874 | 0.887 | 0.704 | **0.787** |
| HOFM-shared-augmented ($m = 4$) | 0.836 | 0.824 | 0.663 | 0.779 |
| HOFM-shared-augmented ($m = 5$) | 0.824 | 0.795 | 0.600 | 0.621 |
| HOFM-shared-simplex ($m = 2$) | 0.716 | 0.865 | **0.721** | 0.701 |
| HOFM-shared-simplex ($m = 3$) | 0.777 | 0.870 | **0.721** | 0.709 |
| HOFM-shared-simplex ($m = 4$) | 0.758 | 0.870 | **0.721** | 0.709 |
| HOFM-shared-simplex ($m = 5$) | 0.722 | 0.869 | **0.721** | 0.709 |
| All-subsets | 0.730 | 0.840 | **0.721** | 0.714 |
| Polynomial network ($m = 2$) | 0.725 | 0.879 | **0.721** | 0.761 |
| Polynomial network ($m = 3$) | 0.789 | 0.853 | 0.719 | 0.696 |
| Polynomial network ($m = 4$) | 0.782 | 0.873 | 0.717 | 0.708 |
| Polynomial network ($m = 5$) | 0.543 | 0.524 | 0.648 | 0.501 |
| Low-rank bilinear regression | 0.855 | 0.694 | 0.611 | 0.718 |

of $\boldsymbol{a}_i$ and $\boldsymbol{b}_j$ and feed this to a HOFM in order to compute a prediction $\hat{y}_{i,j}$. Because HOFMs use feature combinations in $\boldsymbol{x}_{i,j}$, they can learn the weights of feature combinations between $\boldsymbol{a}_i$ and $\boldsymbol{b}_j$. At training time, we need both positive and negative samples. Let us denote the set of positive and negative samples by $\Omega$. Then our training set is composed of $(\boldsymbol{x}_{i,j}, y_{i,j})$ pairs, where $(i, j) \in \Omega$.

**Models compared.**

- *HOFM*: $\hat{y}_{i,j} = \hat{y}_{\text{HOFM}}(\boldsymbol{x}_{i,j})$ as defined in (3) and as originally proposed in [13, 14]. We minimize (4) by alternating minimization of (10) for each degree.

- *HOFM-shared*: $\hat{y}_{i,j} = \sum_{s=1}^{k} \mathcal{A}^{1 \to m}(\boldsymbol{p}_s, \boldsymbol{x}_{i,j}; \boldsymbol{\theta})$. We learn $\boldsymbol{P}$ and $\boldsymbol{\theta}$ using the simple augmented data approach described in Section 5.1 (*HOFM-shared-augmented*). Inspired by SimpleMKL [12], we also report results when learning $\boldsymbol{P}$ and $\boldsymbol{\theta}$ directly by minimizing $\frac{1}{|\Omega|} \sum_{(i,j) \in \Omega} \ell(y_{i,j}, \hat{y}_{i,j}) + \frac{\beta}{2} \|\boldsymbol{P}\|^2$ subject to $\boldsymbol{\theta} \geq 0$ and $\langle \boldsymbol{\theta}, \boldsymbol{1} \rangle = 1$ (*HOFM-shared-simplex*).

- *All-subsets*: $\hat{y}_{i,j} = \sum_{s=1}^{k} \mathcal{S}(\boldsymbol{p}_s, \boldsymbol{x}_{i,j})$. As explained in Section 5.2, this model is equivalent to the HOFM-shared model with $m = n_z(\boldsymbol{x}_{i,j})$ and $\boldsymbol{\theta} = \boldsymbol{1}$.

- *Polynomial network*: $\hat{y}_{i,j} = \sum_{s=1}^{k} (\gamma_s + \langle \boldsymbol{p}_s, \boldsymbol{x}_{i,j} \rangle)^m$. This model can be thought as factorization machine variant that uses a polynomial kernel instead of the ANOVA kernel (c.f., [8, 4, 22]).

- *Low-rank bilinear regression*: $\hat{y}_{i,j} = \boldsymbol{a}_i \boldsymbol{U} \boldsymbol{V}^{\mathrm{T}} \boldsymbol{b}_j$, where $\boldsymbol{U} \in \mathbb{R}^{d_A \times k}$ and $\boldsymbol{V} \in \mathbb{R}^{d_B \times k}$. Such model was shown to work well for link prediction in [9] and [10]. We learn $\boldsymbol{U}$ and $\boldsymbol{V}$ by minimizing $\frac{1}{|\Omega|} \sum_{(i,j) \in \Omega} \ell(y_{i,j}, \hat{y}_{i,j}) + \frac{\beta}{2} (\|\boldsymbol{U}\|^2 + \|\boldsymbol{V}\|^2)$.

**Experimental setup and evaluation.** In this experiment, for all models above, we use CD rather than SG to avoid the tuning of a learning rate hyper-parameter. We set $\ell$ to be the squared loss. Although we omitted it from our notation for clarity, we also fit a bias term for all models. We evaluated the compared models using the area under the ROC curve (AUC), which is the probability that the model correctly ranks a positive sample higher than a negative sample. We split the $n_+$ positive samples into 50% for training and 50% for testing. We sample the same number of negative samples as positive samples for training and use the rest for testing. We chose $\beta$ from $10^{-6}, 10^{-5}, \ldots, 10^6$ by cross-validation and following [9] we empirically set $k = 30$. Throughout our experiments, we initialized the elements of $\boldsymbol{P}$ randomly by $\mathcal{N}(0, 0.01)$.

**Results** are indicated in Table 3. Overall the two best models were HOFM and HOFM-shared-augmented, which achieved the best scores on 3 out of 4 datasets. The two models outperformed low-rank bilinear regression on 3 out 4 datasets, showing the benefit of using higher-order feature combinations. HOFM-shared-augmented achieved similar accuracy to HOFM, despite using a smaller model. Surprisingly, HOFM-shared-simplex did not improve over HOFM-shared-augmented except

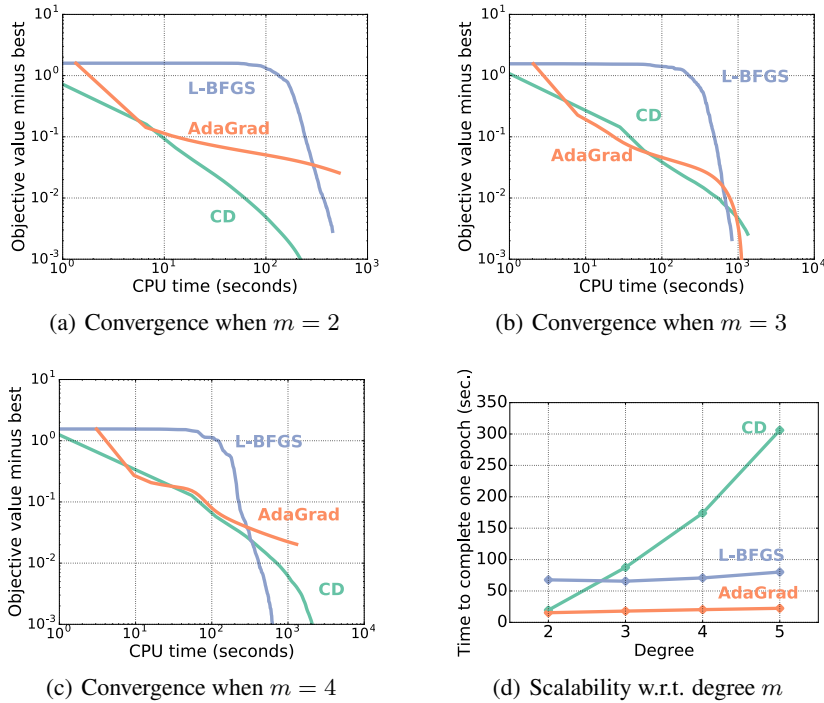

| | |
|---|---|
| (a) Convergence when $m = 2$ | (b) Convergence when $m = 3$ |
| (c) Convergence when $m = 4$ | (d) Scalability w.r.t. degree $m$ |

Figure 1: Solver comparison for minimizing (10) when varying the degree $m$ on the NIPS dataset with $\beta = 0.1$ and $k = 30$. Results on other datasets are in Appendix B.

on the GD dataset. We conclude that our augmented data approach is convenient yet works well in practice. All-subsets and polynomial networks performed worse than HOFM and HOFM-shared-augmented, except on the GD dataset where they were the best. Finally, we observe that HOFM were quite robust to increasing $m$, which is likely a benefit of modeling each degree with a separate matrix.

## 6.2 Solver comparison

We compared AdaGrad [5], L-BFGS and coordinate descent (CD) for minimizing (10) when varying the degree $m$ on the NIPS dataset with $\beta = 0.1$ and $k = 30$. We constructed the data in the same way as explained in the previous section and added $m - 1$ dummy features, resulting in $n = 8,280$ sparse samples of dimension $d = 27,298 + m - 1$. For AdaGrad and L-BFGS, we computed the (stochastic) gradients using Algorithm 2. All solvers used the same initialization.

**Results** are indicated in Figure 1. We see that our CD algorithm performs very well when $m \leq 3$ but starts to deteriorate when $m \geq 4$, in which case L-BFGS becomes advantageous. As shown in Figure 1 d), the cost per epoch of AdaGrad and L-BFGS scales linearly with $m$, a benefit of our DP algorithm for computing the gradient. However, to our surprise, we found that AdaGrad is quite sensitive to the learning rate $\eta$. AdaGrad diverged for $\eta \in \{1, 0.1, 0.01\}$ and the largest value to work well was $\eta = 0.001$. This explains why AdaGrad did not outperform CD despite the lower cost per epoch. In the future, it would be useful to create a CD algorithm with a better dependency on $m$.

## 7 Conclusion and future directions

In this paper, we presented the first training algorithms for HOFMs and introduced new HOFM variants with shared parameters. A popular way to deal with a large number of negative samples is to use an objective function that directly maximize AUC [9, 15]. This is especially easy to do with SG algorithms because we can sample pairs of positive and negative samples from the dataset upon each SG update. We therefore expect the algorithms developed in Section 3 to be especially useful in this setting. Recently, [7] proposed a distributed SG algorithm for training second-order FMs. It should be straightforward to extend this algorithm to HOFMs based on our contributions in Section 3. Finally, it should be possible to integrate Algorithm 1 and 2 into a deep learning framework such as TensorFlow [1], in order to easily compose ANOVA kernels with other layers (e.g., convolutional).

## Footnotes

[1]We include the constant factor for fair later comparison with arbitrary-order HOFMs.

[2]The name comes from the ANOVA decomposition of functions. [20, 19]

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
