[Supplementary Material · supp.pdf]

# Supplementary material

## A   Dataset descriptions

- **NIPS**: co-author graph of authors at the first twelve editions of NIPS, obtained from [17]. For this dataset, as well as the Enzyme dataset below, we have $\boldsymbol{A} = \boldsymbol{B}$. The co-author graph comprises $n_A = n_B = 2,037$ authors represented by bag-of-words vectors of dimension $d_A = d_B = 13,649$ (words used by authors in their publications). The number of positive samples is $n_+ = 4,140$.

- **Enzyme**: metabolic network obtained from [21]. The network comprises $n_A = n_B = 668$ enzymes represented by three sets of features: a $157$-dimensional vector of phylogenetic information, a $145$-dimensional vector of gene expression information and a $23$-dimensional vector of gene location information. We concatenate the three sets of information to form feature vectors of dimension $d_A = d_B = 325$. Original enzyme similarity scores are between $0$ and $1$. We binarize the scores using $0.95$ as threshold. The resulting number of positive samples is $n_+ = 2,994$.

- **GD**: human gene-disease association data obtained from [10]. The bipartite graph is comprised of $n_A = 3,209$ diseases and $n_B = 12,331$ genes. We represent each disease using a vector of $d_A = 3,209$ dimensions, whose elements are similarity scores obtained from MimMiner. The study [10] also used bag-of-words vectors describing each disease but we found these to not help improve performance both for baselines and proposed methods. We represent each gene using a vector of $d_B = 25,275$ features, which are the concatenation of $12,331$ similarity scores obtained from HumanNet and $12,944$ gene-phenotype associations from 8 other species. See [10] for a detailed description of the features. The number of positive samples is $n_+ = 3,954$.

- **Movielens 100K**: recommender system data obtained from [6]. The bipartite graph is comprised of $n_A = 943$ users and $n_B = 1,682$ movies. For users, we convert age, gender, occupation and living area (first digit of zipcode) to a binary vector using a one-hot encoding. For movies, we use the release year and genres. The resulting vectors are of dimension $d_A = 49$ and $d_B = 29$, respectively. Original ratings are between $1$ and $5$. We binarize the ratings using $5$ as threshold, resulting in $n_+ = 21,201$ positive samples.

## B   Additional experiments

### B.1   Solver comparison

We also compared AdaGrad, L-BFGS and coordinate descent (CD) on the Enzyme, Gene-Disease (GD) and Movielens 100K datasets. Results are indicated in Figure 2, 3 and 4, respectively.

### B.2   Recommender system experiments

As we explained in Section 5.2, the all-subsets kernel can be a good choice if the number of non-zero elements per sample is small. To verify this assumption, we ran experiments on two recommender system tasks: Movielens 1M and Last.fm. We used the exact same setting as in [4, Section 9.3]. For each rating $y_i$, the corresponding $\boldsymbol{x}_i$ was set to the concatenation of the one-hot encodings of the user and item indices. We compared the following models:

- FM: $\hat{y}_i = \langle \boldsymbol{w}, \boldsymbol{x}_i \rangle + \sum_{s=1}^{k} \mathcal{A}^2(\boldsymbol{p}_s, \boldsymbol{x}_i)$
- FM-augmented: $\hat{y}_i = \sum_{s=1}^{k} \mathcal{A}^2(\boldsymbol{p}_s, \tilde{\boldsymbol{x}}_i)$ where $\tilde{\boldsymbol{x}}_i^{\mathrm{T}} = [1, \boldsymbol{x}_i^{\mathrm{T}}]$
- All-subsets: $\hat{y}_i = \sum_{s=1}^{k} \mathcal{S}(\boldsymbol{p}_s, \boldsymbol{x}_i)$
- Polynomial networks: $\hat{y}_i = \tilde{\boldsymbol{x}}_i \boldsymbol{U} \boldsymbol{V}^{\mathrm{T}} \tilde{\boldsymbol{x}}_i$ (c.f. [4] for more details)

Results are indicated in Figure 5. We see that All-subsets performs relatively well on these tasks.

(a) $m = 2$      (b) $m = 3$      (c) $m = 4$

Figure 2: Solver comparison for minimizing (10) on the Enzyme dataset. We set $\beta$ to the values with best test-set performance, which were $\beta = 0.1$, $\beta = 0.1$ and $\beta = 0.01$, respectively. We set $k = 30$.

(a) $m = 2$      (b) $m = 3$      (c) $m = 4$

Figure 3: Solver comparison for minimizing (10) on the GD dataset. We set $\beta$ to the values with best test-set performance, which were $\beta = 0.01$, $\beta = 0.01$ and $\beta = 0.0001$, respectively. We set $k = 30$.

(a) $m = 2$      (b) $m = 3$      (c) $m = 4$

Figure 4: Solver comparison for minimizing (10) on the Movielens 100K dataset. We set $\beta$ to the values with best test-set performance, which were $\beta = 10^{-3}$, $\beta = 10^{-4}$ and $\beta = 10^{-6}$, respectively. We set $k = 30$.

(a) Movielens 1M      (b) Last.FM

Figure 5: Model comparison on two recommender system datasets.

# C   Reverse-mode differentiation on the alternative recursion

We now describe how to apply reverse-mode differentiation to the alternative recursion (11) in order to compute the entire gradient efficiently. Let us introduce the shorthands $a_t := \mathcal{A}^t(\boldsymbol{p}, \boldsymbol{x})$ and $d_t := \mathcal{D}^t(\boldsymbol{p}, \boldsymbol{x})$. We can then write the recursion as

$$a_m = \frac{1}{m} \sum_{t=1}^{m} (-1)^{t+1} a_{m-t} d_t.$$

For concreteness, let us illustrate the recursion for $m = 3$. We have

$$a_1 = a_0 d_1, \quad a_2 = \frac{1}{2}(a_1 d_1 - a_0 d_2) \quad \text{and} \quad a_3 = \frac{1}{3}(a_2 d_1 - a_1 d_2 + a_0 d_3).$$

We see that $a_2$ influences $a_3$, and $a_1$ influences $a_2$ and $a_3$. Likewise, $d_3$ influences $a_3$, $d_2$ influences $a_2$ and $a_3$, and $d_1$ influences $a_1$, $a_2$ and $a_3$. Let us denote the adjoints $\tilde{a}_t := \frac{\partial a_m}{\partial a_t}$ and $\tilde{d}_t := \frac{\partial a_m}{\partial d_t}$. For general $m$, summing over quantities that influences $a_t$ and $d_t$, we obtain

$$\tilde{a}_t = \sum_{s=t+1}^{m} \frac{(-1)^{s-t+1}}{s} \tilde{a}_s d_{s-t} \quad \text{and} \quad \tilde{d}_t = (-1)^{t+1} \sum_{s=t}^{m} \frac{1}{s} \tilde{a}_s a_{s-t}.$$

Let us denote the adjoint of $p_j$ by $\tilde{p}_j := \frac{\partial a_m}{\partial p_j}$. We know that $p_j$ directly influences only $d_1, \ldots, d_m$ and therefore

$$\tilde{p}_j = \sum_{t=1}^{m} \frac{\partial a_m}{\partial d_m} \frac{\partial d_m}{\partial p_j} = \sum_{t=1}^{m} \tilde{d}_t t p_j^{t-1} x_j^t.$$

Assuming that $d_1, \ldots, d_m$ and $a_1, \ldots, a_m$ have been previously computed, which takes $O(dm + m^2)$, the procedure for computing the gradient can be summarized as follows:

1. Initialize $\tilde{a}_m = 1$,
2. Compute $\tilde{a}_{m-1}, \ldots, \tilde{a}_1$ (in that order),
3. Compute $\tilde{d}_m, \ldots, \tilde{d}_1$,
4. Compute $\nabla \mathcal{A}^m(\boldsymbol{p}, \boldsymbol{x}) = [\tilde{p}_1, \ldots, \tilde{p}_d]^{\mathrm{T}}$.

Steps 2 and 4 both take $O(m^2)$ and step 4 takes $O(dm)$ so the total cost is $O(dm + m^2)$. We can improve the complexity of step 4 as follows. We can rewrite $\nabla \mathcal{A}^m(\boldsymbol{p}, \boldsymbol{x})$ in matrix notation:

$$\nabla \mathcal{A}^m(\boldsymbol{p}, \boldsymbol{x}) = \left( \begin{bmatrix} 1 & p_1 x_1 & (p_1 x_1)^2 & \ldots & (p_1 x_1)^{m-1} \\ 1 & p_2 x_2 & (p_2 x_2)^2 & \ldots & (p_2 x_2)^{m-1} \\ \vdots & \vdots & \vdots & \ddots & \vdots \\ 1 & p_d x_d & (p_d x_d)^2 & \ldots & (p_d x_d)^{m-1} \end{bmatrix} \begin{bmatrix} \tilde{d}_1 \\ 2\tilde{d}_2 \\ \vdots \\ m\tilde{d}_m \end{bmatrix} \right) \circ \boldsymbol{x}.$$

The left matrix is called a Vandermonde matrix. The product between a $d \times m$ Vandermonde matrix and a $m$-dimensional vector can be computed using the Moenck-Borodin algorithm (an algorithm similar to the FFT), in $O(r \log^2 l)$, where $r = \max(d, m)$ and $l = \min(d, m)$ [11]. Since $m \leq d$, the cost of step 4 can therefore be reduced to $O(d \log^2 m)$.