[Reviews · NeurIPS 2016]

Reviewer 1

Summary

The paper extends a model class called Factorization Machines (FMs) by making it feasible to go beyond the pairwise feature interactions typically modeled when using FMs and allowing to go for higher order interactions. Learning traditional higher order FMs is a challenge due to the costs in computing predictions and gradients. The paper The authors address this issue by reformulating the prediction function using the ANOVA kernel. A linear time algorithm for evaluating the ANOVA kernel and computing its gradient is presented which makes it feasible to train FMs using higher order interactions.

Qualitative Assessment

This is an excellent paper, well written and with a solid contribution. One question one might ask is on which situations, modelling higher order interactions does lead to improve predictive performance. From the experimental results it seems that going for 3-wise interactions seems to work best for the analyzed dataset with a small improvement over pairwise interactions. Another point that would be interesting to see is how the proposed algorithm compares with the standard FM implementation (e.g. libFM) in terms of prediction and learning runtime . Typos: line 263: convulational --> convolutional

Confidence in this Review

2-Confident (read it all; understood it all reasonably well)


Reviewer 2

Summary

The authors exploit the connection between higher order factorization machines and ANOVA kernels to present a dynamic programming approach to efficiently computing the gradient of HOFMs in linear time, which makes fitting HOFMs feasible for the first time. They suggest using a coordinate descent algorithm and show that it is competitive (to LBFGS and Adagrad) when the order is less than three. They use the same connections to ANOVA kernels to introduce a variant of HOFMs with many fewer parameters by sharing the coefficients between the different orders of interactions, and to show how to learn using the all-subsets kernel. They provide experimental results showing that HOFMs outperform second-order factorization machines and the state-of-the-art bilinear regression models for the task of link prediction on four datasets. Comments: the tensor machines model of Yang and Gittens is relevant prior work

Qualitative Assessment

The work is interesting because it provides a practical scheme for fitting higher order factorization machines. I know quadratic FMs are very popular for recommendation tasks, so it's good to see work that makes it possible to see if higher order versions are useful. However, the paper is so dense with the minutiae of calculation that it is difficult to see it being read and implemented widely (if the authors were to write a package for this, it might be used). I would have preferred to see the CD section dropped, and more time spent on explaining the reverse differentiation used to compute the gradients. Also, I have computed the analogues of the efficient gradient evaluation formulas for the quadratic FMs for up to fifth order HOFMs, so it's not clear that all this machinery is necessary. When does one need higher order polynomial interactions?

Confidence in this Review

2-Confident (read it all; understood it all reasonably well)


Reviewer 3

Summary

This work presents an efficient algorithms for training higher-order FMs. This work also presents new variants of HOFMs with shared parameters, which greatly reduce model size and prediction times while maintaining similar accuracy. This work demonstrates the proposed approaches on four different link prediction tasks.

Qualitative Assessment

1. There is lack experimental result comparing to the standard FMs. It can demonstrate that if it is necessary to use higher-order FMs instead of 2-order FMs. 2. In line 253, the sentence, i.e. "this algorithm may outperform CD for larger m", is not rigorous. All conclusions should be derived from the experimental results. 3. Only AUC is reported as evaluation metrics; what about other metrics like Recall, Precision, and F-score?

Confidence in this Review

2-Confident (read it all; understood it all reasonably well)


Reviewer 4

Summary

The authors propose a learning algorithm for so called higher order FMs (HOFMs) based on ANOVA kernels and the use of HOFMs with shared parameters. A backpropagation like algorithm is used for training.

Qualitative Assessment

It is an interesting, reasoned and promising approach. But there a few issues which I would like to have clarified in the rebuttal to accept the paper. 1. The idea of the paper seems to strongly rely on the paper "Polynomial Networks and Factorization Machines: New Insights and Efficient Training Algorithms" by Blondel et al., where ANOVA kernels have already been used. Can you explain in more detail the difference and contributions in comparison to this paper? 2. There has been extensive work on tensor factorizations which I am missing here. E.g., many papers by Tamara Kolda. I'm wondering why such approaches cannot be applied to the given problem or why it is not better to adapt them to HOFMs. Such comparisons to state-of-the art tensor factorizations are accordingly also missing in the experiments. Are there reason for this? If so, please highlight them. 3. The experiments are not very extensive. There is just one quite specific setting based on the prediction of presence or absence of links in graphs which isn't explained in much detail. Moreover, your method does not consistently perform better than the baselines. Not just for GD, also the performance differences for the other data sets are low. For NIPS the low-rank bilinear regression comes close to the value attained by a HOFM. For Enzymes and Movielens a Polynomial Network comes very close. It should be noted that the values vary quite a lot for different orders m, even in a nonmonotonic way. For example, the difference between a Polynomial Network with m=3 and both m=2, m=4 is bigger than the difference between a Polynomial Network with m=2 (or m=4) and the best value for a HOFM. This makes me wonder whether those values didn't just happen by chance. You also seem to have tried more settings for HOFMs than for the baselines, so it could be man-made overfitting. Also the data sets are rather small which could raise scalability questions. More detailed experiments would be better in my opinion. At least these issues should be explained.

Confidence in this Review

3-Expert (read the paper in detail, know the area, quite certain of my opinion)


Reviewer 5

Summary

This work studied efficient training algorithms for higher-order factorization machines (HOFMs). The link between FMs and the ANOVA kernel was used. It first proposed dynamic programming (DP) algorithms for evaluating the ANOVA kernel and its gradient. Then stochastic gradient and coordinate descent algorithms for HOFMs were presented. Variants of the ANOVA kernel with shared parameters were also discussed.

Qualitative Assessment

This work relied heavily on Blondel et al.’s recent work [4], including the connection of FM with the ANOVA kernel; the multi-linearity of ANOVA kernel which triggered the DP algorithms for kernel evaluation; and the coordinate descent (CD) algorithm for solving the optimization problem. The major focus of this paper is the extension of the CD from m=3 to arbitrary order. From this view point, the contribution of this work seems limited. Besides, according to the experiment results, the CD algorithm did not show obvious advantage over other solvers like L-BFGS. The results also showed that HOFM with orders higher than 3 did not outperform lower-order FMs, which may make the HOFM models and their efficient training algorithms less important. In Section 3, a stochastic gradient algorithm was presented, which has been claimed as one of the contributions in Line 27. However its performance was not shown in the experiment part. The polynomial network method in [4] was compared in Table 3. How about the FM method in [4]? Is it the same as HOFM with the order equals 2 and 3?

Confidence in this Review

1-Less confident (might not have understood significant parts)